# Toxin Accumulation, Distribution, and Sources of Toxic Xanthid Crabs

**DOI:** 10.3390/toxins17050228

**Published:** 2025-05-05

**Authors:** Yuchengmin Zhang, Hongchen Zhu, Tomohiro Takatani, Osamu Arakawa

**Affiliations:** 1Jiangsu Key Laboratory of Marine Bioresources and Environment/Jiangsu Key Laboratory of Marine Biotechnology, Jiangsu Ocean University, Lianyungang 222005, China; 2Co-Innovation Center of Jiangsu Marine Bio-Industry Technology, Jiangsu Ocean University, Lianyungang 222005, China; 3Graduate School of Fisheries and Environmental Sciences, Nagasaki University, 1-14, Bunkyo-machi, Nagasaki 852-8521, Japan; zhc957286316@hotmail.com; 4Graduate School of Integrated Science and Technology, Nagasaki University, 1-14, Bunkyo-machi, Nagasaki 852-8521, Japan; taka@nagasaki-u.ac.jp (T.T.); arakawa@nagasaki-u.ac.jp (O.A.)

**Keywords:** toxic xanthid crabs, tetrodotoxin, paralytic shellfish toxins, palytoxin, geographic distribution, origin

## Abstract

Several species of crabs from the Xanthidae family are recognized as dangerous marine organisms due to their potent neurotoxins, including paralytic shellfish toxin (PST), tetrodotoxin (TTX), and palytoxin (PLTX). However, the mechanisms of toxin accumulation and transport and the origin of these toxins in toxic xanthid crabs remain unknown. The identification of toxic crab species, their toxicity and toxin composition, and toxin profiles have been studied thus far. To date, more than ten species of xanthid crabs have been confirmed to possess toxins. Recently, several new studies on crabs, including the geographic distribution of toxin profiles and the ecological role of crabs, have been reported. Therefore, this review provides a summary of global research on toxic xanthid crabs, containing new findings and hypotheses on the toxification in and the origins of these crabs. Furthermore, the challenges and future perspectives in this field are also discussed.

## 1. Introduction

Xanthid crabs, belonging to the Xanthidae family, are a diverse group of marine crustaceans that are known for their vibrant coloration and their significant role in coastal ecosystems. Various incidents caused by toxic xanthid crabs have been reported in several parts of the world over the years [1,2]. Research on toxic xanthid crabs began in the 1960s in Japan, followed by screening studies in Australia, Taiwan, Vietnam, and the Philippines, where the toxicity and toxin components of these crabs were identified [3,4,5,6,7,8]. The screening studies were firstly identified by mouse bioassay (MBA). Recently, with the development in detection equipment and methods, high-performance liquid chromatography (HPLC) and liquid chromatography–tandem mass spectrometry (LC-MS/MS) have become commonly used analytical tools [9]. Furthermore, other detection methods have also been applied to the detection of biotoxins, such as enzyme-linked immunosorbent assay (ELISA), biosensors, and flow injection microfluidic immunoassay systems (FI-IA system) [10,11,12,13,14,15,16,17].

The most well-known toxic xanthid crab species are *Atergatis floridus* and *Zosimus aeneus*, which have extremely high toxicity, and their toxin profiles from different regions showed the presence of paralytic shellfish toxin (PST) and tetrodotoxin (TTX) [18,19]. Furthermore, in 1986–1999, several xanthid crabs were found to possess palytoxin (PLTX) and/or palytoxin-like toxins [20]. Due to the individual and geographic differences in toxin contents and profiles reported by many studies [9], these toxins are considered to have an exogenous origin. However, to date, the mechanism of toxification, including toxin accumulation, transport, and metabolism, in toxic xanthid crabs is not yet fully understood. This review aims to summarize the previous studies on the toxin profiles and distribution of toxic xanthid crabs, along with recent findings on potential source organisms and their ecological roles.

## 2. Xanthid Crabs

Xanthid crabs of the Xanthidae family comprise 12 subfamilies, 53 genera, and more than 600 species, and are widely distributed in tropical and subtropical coastal regions and intertidal areas such as the Indo-Pacific, Mediterranean, and Atlantic regions [21]. Recent studies have identified several new species within this family, indicating that they are a large group of crustaceans [22,23,24]. Their hard exoskeleton can protect them from predators, while their strong claws can be used for foraging and defense against threats. Similar to other crabs, they are usually omnivores, with a diverse diet that includes algae, sponges, mollusks, and ascidians [25]. The reproductive cycle of xanthid crabs is generally seasonal, the female species can produce thousands of eggs, and the eggs may develop into planktonic larvae [26]. As they grow older, their life cycle always involves molting, just like other crustaceans, to change the old carapace to a new one [26,27,28]. Although most xanthid crabs are non-toxic, several species within the Xanthidae family have shown confirmed toxicity on screening. Most toxic crabs are easily identified by their rounded and hard carapace with colorful patterns or lines, which are commonly used as a danger signal to warn predators of their toxicity [29]. Interestingly, the old carapaces are non-toxic even in toxic species [2].

### 2.1. First Three Species Identified as Toxic

A previous Japanese study conducted in the 1960s investigated approximately 1000 crab specimens from 72 species belonging to 8 families and found that 3 species belonging to the Xanthidae family, *Z. aeneus*, *A. floridus*, and *Platypodia granulosa*, were toxic [3].

#### 2.1.1. *Zosimus aeneus*

*Z. aeneus* was first described in the 10th edition of *Systema Naturae*, published by Carl Linnaeus in 1758 [30], commonly known as the “devil crab.” It is a well-known species known for its bright colors and beautiful patterns. Its carapace, claws, and legs exhibit a unique pattern of red, purple, or deep blue patches on a pale brown or cream-colored surface, and usually, its abdomen is cream-colored (Figure 1A). Its carapace has deep grooves, while the tips of its walking legs are characterized by a small fuzz [30]. *Z. aeneus* can reach a size of 6–9 cm and is the largest species among the three toxic crabs (*Z. aeneus*, *A. floridus*, and *P. granulosa*). This species inhabits live coral reefs across the Indo-Pacific region, with its distribution spanning from the Red Sea to the Ryukyu Islands and Hawaii. *Z. aeneus* is considered a critical species for toxicity research and has been involved in several human poisoning incidents.

#### 2.1.2. *Atergatis floridus*

*A. floridus* is distinguished by its smooth surface and round carapace. It is called “sube-sube-manju” in Japan because the shape of its carapace is like a smooth steamed bun. It has a unique greenish-to-greenish-blue–brown color or a dark-brown color, with a lace-like pattern of white or yellow lines (Figure 1B). These unique characteristics make it different from other crab species [31,32]. *A. floridus* is distributed more widely than *Z. aeneus*. Usually, it is found in tropical reef ecosystems and along rocky shores of the Indian Ocean, the Western Pacific region, and the southern coast of Taiwan. *A. floridus*, found in many regions, contains extremely high levels of toxins like *Z. aeneus*.

**Table 1 toxins-17-00228-t001:** Summary of the identification of toxic xanthid crabs.

Species	Toxic Specimens/Tested Specimens	Toxin Profiles	Toxin Contents	Methods of Determination	Location	Reference
*Zosimus aeneus*	81/112	——	NM	MBA^3^	Ryukyu and Amami Islands, Japan	[3]
	77/107	——	ND-1580 MU/g	MBA	Kyonoura, Amami-Oshima Island, Ishigaki Island, and Tokunoshima Island, Japan	[4]
	24/24	——	30–1260 MU/g	MBA	Ishigaki Island, Japan	[5]
	5/28	——	ND-30 MU/g	MBA	Marcus Island, Japan	
	0/38	——	ND	MBA	Espiritu Santo, the Republic of Vanuatu	
	0/2	——	ND	MBA	American Samoa	
	9/9	——	30–260 MU/g	MBA	Rangiroa, Tuamotu Island	
	101/102	——	ND-16500 MU/g	MBA	Ishigaki Island, Japan	[33]
	6/6	——	2–220 MU/g	MBA	Hachijo Island and Bonin Islands, Japan	
	11/11	——	26–3000 MU/g	MBA	Cebu Island, the Philippines	
	18/18	PST	2.2–33.8 MU/g	MBA, TLC	Fiji Island, Fiji	[34]
	50/69	PST and/or TTX	ND-259 MU/g	MBA, HPLC	Negros Island, the Philippines	[35]
	53/75	PST and/or TTX	ND-2300 MU	MBA, HPLC	Lanyu, Wanlitung, and Hsiaoliuchiu, Taiwan	[7]
	14/15	PST and/or TTX	ND-1258 MU	MBA, HPLC, LC-MS, GC-MS	Kenting National Park, Taiwan	[36]
	9/16	TTX	ND-11 MU/g	MBA, HPLC, LC-MS	Tokara Islands, Japan	[37]
	52/53	PST and/or TTX	ND-14700 MU/g (appendages)	HPLC, LC-MS	Ishigaki Island, Japan	[9]
*Atergatis floridus*	102/144	——	NM	MBA	Ryukyu Islands, Amami Islands, and the mainland of Japan, Japan	[3]
	66/96	——	NM	MBA	Nagasaki, Kaimon, Tosa, Minabe, Nagai, Awa-Shirahama, Hachijo Island, and Oshima Island, Japan	[4]
	35/35	PST	3.4–717 MU/g	MBA, TLC	Fiji Island, Fiji	[34]
	19/21	PST and/or TTX	ND-1300 MU/g	MBA, TLC, HPLC	Ishigaki Island, Japan	[38]
	25/25	PST	2.3–480 MU/g	MBA, HPLC	Negros Island, the Philippines	[35]
	65/109	PST	ND-108 MU/g	MBA, TLC	The Great Barrier Reef, Australia	[8]
	8/8	PST and/or TTX	63–424 MU	MBA, TLC, HPLC	Keelung, Taiwan	[39]
	32/32	TTX	3–237 MU/g (muscle of cheliped)	MBA	Kanagawa and Wakayama, Japan	[40]
	15/15	PST and/or TTX	88 ± 40–1257 ± 607 MU/g (appendages Av. ± S.D.)	MBA, HPLC, GC-MS	Ishigaki Island, Japan	[41]
	5/5	PST and/or TTX	221 ± 189 MU/g (appendages Av. ± S.D.)	MBA, HPLC, GC-MS	Cebu, the Philippines	[41]
	9/9	TTX	2.73–206.64 MU/g	MBA, HPLC, GC-MS	Nagasaki, Japan	[42]
	15/15	TTX	0.21–116.1 MU/g (appendages)	MBA, HPLC, GC-MS	Nagasaki, Japan	[42]
*Lophozozymus pictor*	4/4	PLTX	200–10,000 MU/g (viscera) 180–3000 MU/g (gill)	TLC, HPLC, FAB-MS	Negros Island, the Philippines	[43]
	——	PLTX	5000 MU/g (extract)	MBA, TLC, HPLC	Sentosa Island, Singapore	[44]
	21/21	——	7840 ± 2254 MU/g (gut Av. ± S.D.)	MBA	Sentosa Island, Singapore	[45]
	15/15	PST	386–3874 MU	MBA, TLC, HPLC	Keelung, Taiwan	[46]
*Platypodia granulosa*	12/38	——	NM	MBA	Ryukyu Islands, Japan	[3]
	11/35	——	NM	MBA	Miyako Island, and Ryukyu Islands, Japan	[4]
*Atergatopsis germaini*	17/17	PST	472–9045 MU	MBA, TLC, HPLC	Keelung, Taiwan	[47]
*Atergatis integerrimus*	1/4	TTX	ND-2 MU/g	MBA	Negros Island, the Philippines	[35]
*Actaeodes tomentosus*	7/15	TTX	ND-40 MU	MBA, HPLC, LC-MS, GC-MS	Kenting National Park, Taiwan	[36]
*Carpilius maculatus*	1/2	——	ND-2 MU/g	MBA	Negros Island, the Philippines	[35]
*Etisus rhynchophorus*	1/2	——	ND-2 MU/g	MBA	Negros Island, the Philippines	[35]
*Etisus splendidus*	1/4	——	ND-2 MU/g	MBA	Negros Island, the Philippines	[35]
*Eriphia sebana*	35/75	PST	ND-9 MU/g	MBA, TLC	The Great Barrier Reef, Australia	[48]
*Demania reynaudi*	7/7	PST and/or TTX	217–564 MU	MBA, TLC, HPLC	Keelung, Taiwan	[39]
	1/1	PLTX-like toxin	800 MU (two legs)	MBA, HPLC	Negros Island, the Philippines	[49]
*Demania cultripes*	7/7	PST and/or TTX	5.6–52.1 MU/g (viscera)	MBA, HPLC, GC-MS	Cebu Island, the Philippines	[50]
*Demania alcalai*	2/2	PLTX	4000–5400 (viscera) 2400–16,000 (gill)	TLC, HPLC, FAB-MS	Negros Island, the Philippines	[43]
*Leptodius sanguirreus*	3/3	——	0.3–1.3 MU/g	MBA	Heron Island, Australia	[51]
*Pilodius areolatus*	4/10	——	1.5 MU/g (extract)	MBA	Heron Island, Australia	[51]
*Phymodius angolalos*	6/9	——	1.6–8.9 MU/g (three extracts)	MBA	Heron Island, Australia	[51]
*Platypodiella spectabilis*	3/3	PLTX	20–50 HU/g	Hemolysis, HPLC	Santa Marta, Colombia	[52,53]
*Xanthias lividu*	9/9	PST and/or TTX	118–430 MU	MBA, HPLC	Lanyu and Hsiaoliuchiu, Taiwan	[54]
	5/5	PST and/or TTX	11–118 MU	MBA, HPLC, LC-MS, GC-MS	Kenting National Park, Taiwan	[36]

——: No toxins were detected or mentioned. NM: not mentioned. MU: mouse unit. ND: not detected. MBA: mouse bioassay. HU: hemolytic unit.

#### 2.1.3. *Platypodia granulosa*

*P. granulosa* is one of the few poisonous species with a body width ranging from 4 to 6 cm. *P. granulosa* displays a subdued chocolate-brown coloration on both its body and its claws, with a pair of red eyes distinguishing it from other colored xanthid species (Figure 1C). Its carapace is covered with large, rounded bumps, giving it a distinctive texture. The sides of the body are smooth with regular notches, often highlighted by pale bars, making the crabs seem like a pie [55,56].

### 2.2. Species Subsequently Identified as Toxic

Following this study, the screening of toxic crabs was continued in Japan and was also conducted in the Philippines, Australia, Taiwan, and other parts of the world [35,51,57]. The findings from these studies indicated that several species of xanthid crabs all over the world are considered to be toxic, and they are summarized in Table 1. With the toxin incidents and the investigations of toxic crabs in various countries and regions, over 10 species of the Xanthidae family have been reported in the past half-decade. Toxic species identified during this period include *Lophozozymus pictor*, *Atergatopsis germaini*, and *Xanthia lividus* in Taiwan and Australia [46,51]; *Demania* species such as *Demania reynaudi* in the Philippines and Taiwan [39,49]; and *Demania cultripes* and *Demania alcalai* in Vietnam and the Philippines [43,50]. In Australia, more types of toxic xanthid crabs were identified in 1988; although their toxin profiles cannot be confirmed and their toxicities are low, these species of xanthid crabs can also be dangerous [51]. From 2008 to 2009, a screening study of marine Brachyuran crabs in Indian waters investigated a total of 990 species belonging to 281 genera and 36 families. Among these, eight families were found to have toxic species. Notably, the majority of the toxic crabs identified belong to the Xanthidae family [29,56]. A brief introduction of *P. granulosa*, *L. pictor*, and *Demania* species is presented below.

#### 2.2.1. *Lophozozymus pictor*

*L. pictor* is another famous toxic xanthid crab. This species has an 8–10 cm body width and has special colors and grooves on its carapace. It is mostly red in color, with several white mosaic and black spots on the tip of its claws [45,58].

#### 2.2.2. *Demania* Species

*Demania* is a genus of crabs from the Xanthidae family, commonly found in the intertidal zone. Several species within this genus are known to contain toxins. Their carapace width usually reaches about 50 mm and has characteristic patterns. Moreover, their carapaces are striking, exhibiting mainly brown color with some colorful features, such as maroon, orange, beige, and white markings. Almost all of the legs have orange and white bands or lines [59].

## 3. Toxin Profiles and Contents in Toxic Crabs

The investigations of toxic xanthid crabs from around the world have found that they usually contain PSTs and TTX along with their analogs [19,60,61]. In addition, some studies have indicated that they can also accumulate PLTXs and PLTX-like toxins [62]. PSTs and TTX are close in molecular weight, lethal to humans, and cause muscle paralysis by specifically blocking voltage-gated sodium (Na_v_) channels, while PLTXs and PLTX-like toxins can block the Na^+^/K^+^-ATPase channel [63]. Table 1 shows the toxin investigations of toxic xanthid crabs summarized in this review from 1969 to 2023.

### 3.1. Paralytic Shellfish Toxins (PSTs)

PSTs are a group of neurotoxins produced by toxic dinoflagellates and cyanobacteria and are involved in the toxification of bivalves [64]. Around 1964, a paralytic shellfish toxin from *Saxidomus giganteus* was named saxitoxin, and its structure was determined by Schantz et al. in 1975 [65]. It has a trialkyl tetrahydropurine as its basic structure, featuring a 3,4-propino-perhydropurine tricyclic system. NH_2_ groups at positions 2 and 8 of the purine ring form two stable guanidinium moieties, contributing to its high polarity and hydrophilicity [65]. In addition, many different organisms have been found to possess PSTs, including frogs, ascidians, pufferfish, and crabs [66,67]. The PST chemical structures were deciphered using an X-ray diffraction analysis in 1975. To date, more than 50 analogs of PSTs have been identified [68]. Figure 2 shows the PST analogs usually detected in the toxic crabs. They are commonly classified into gonyautoxin groups (GTXs), including GTX 1–4 and dcGTX2, 3, and saxitoxin groups (STXs), including STX, neoSTX, dcSTX, and dcneoSTX. Two specific PST analogs, carbamoyl-*N*-hydroxysaxitoxin (hySTX) and carbamoyl-*N*-hydroxyneosaxitoxin (hyneoSTX), were first isolated from xanthid crabs and have not yet been detected in other organisms [69]. In addition, another specific analog, 11-saxitoxinethanoic acid (SEA), has been reported to be identified in *A. floridus* living on the Pacific coast of Shikoku Island [70]. This STX analogue is unique for its carbon–carbon bond at the C11 position (Figure 2). Different analogs have different toxicities. Commonly, the most toxic component is STX, followed by neosaxitoxin (neoSTX) and gonyautoxins 1 and 3 (GTX 1 and 3) [71]. Moreover, the transformation of GTXs in toxic crabs has been reported. Kotaki et al. [72] isolated two PST-transforming bacteria, *Pseudomonas* sp. and *Vibrio* sp., in *A. floridus* that can transform GTX2, 3 into STX. Arakawa et al. [73] demonstrated that *A. floridus* can accumulate GTXs with external exposure by converting them into STXs. In this study, 27 crab specimens were divided into nine groups. Four groups received injections of a GTX mixture. The remaining four groups were fed a toxic diet once daily for 3, 6, 9, and 12 days. The results showed that *A. floridus* quickly converted GTXs into STXs under both the injection and feeding conditions. Recently, Zhang et al. [9] investigated the toxin profiles of *Z. aeneus* by analyzing both its appendages and its stomach contents. Their findings revealed that, while the stomach contents contained lower toxin concentrations compared to the appendages, the composition of PSTs in the stomach differed significantly. GTXs were predominant in many stomach contents of this specimen. Thus, the authors hypothesized that xanthid crabs may ingest GTX-dominated toxins through their toxic prey and change these toxins into STXs for accumulation within their tissues. Although PST-transforming enzymes have been identified in various organisms, the mechanisms of PST transformation in toxic crabs remain unclear [74].

### 3.2. Tetrodotoxin (TTX)

TTX is known as a potent neurotoxin that was first identified in pufferfish [75,76,77]. Subsequently, TTX and TTX analogs have been identified in various organisms such as octopus [78], newt [79], frog [80], and crab [19]. TTX is a crystalline compound that consists of a guanidinium moiety attached to a highly oxygenated carbon skeleton, which contains a 2,4-dioxaadamantane fragment with five hydroxyl groups [81]. To date, approximately 30 analogs of TTX have been identified in both marine and terrestrial animals. Some of these analogs exhibit significant toxicity, especially 11-oxoTTX, which is known to be as toxic as or even more toxic than TTX [82]. Figure 3 shows the TTX and TTX analogs commonly detected in toxic xanthid crabs. They are usually distinguished by differences in the structures of R3 and R4, except for 4-*epi*TTX, whose R1 and R2 structures differ from those of TTX. In addition, TTX, 4-*epi*TTX, and 4,9-anhydroTTX are usually found together due to the chemical equilibrium that exists among them [83]. 11-oxoTTX and 11-norTTX-6(*R*)-ol were identified from *A. floridus*, and 11-oxoTTX was present in almost the same amount as TTX, indicating that oxidation might be more advanced in *A. floridus* than in other TTX-bearing animals [61]. In contrast, 11-oxoTTX, deoxyTTX, 4-*epi*TTX, and 4,9-anhydroTTX have been isolated in *Z. aeneus* on Yoshihara Reef, Ishigaki Island [9], and 11-*nor*TTX-6(*S*)-ol has been identified in *A. floridus* from Nagasaki [42].

### 3.3. Palytoxins (PLTXs) and PLTX-like Compounds

Although most toxic xanthid crabs are known to possess TTX and PSTs, a different toxin was identified in three species of xanthid crabs on Negros Island, the Philippines, from 1986 to 1988. The species *D. alcalai*, *L. pictor*, and *D. reynaudii* were found to contain PLTXs and/or PLTX-like compounds [43,49]. In one reported case, a victim experienced restlessness, muscle cramps, and vomiting after eating about a quarter of a crab and died the next day. The toxin responsible was identified as PLTXs and PLTX-like compounds, which are known to be one of the most potent marine toxins. Subsequent studies also confirmed the presence of PLTXs in *L. pictor* from Singapore. Furthermore, PLTXs were also identified in the xanthid crab *Platypodiella spectabilis* from Colombia [52].

PLTXs is a group with high-molecular-weight components with complex structures, and their molecular weight is approximately 2680 (Figure 4). PLTXs were first isolated from the zoanthid *Palythoa toxica* in Hawaii [84]. Similar to TTX and PSTs, various marine organisms contain PLTXs and/or PLTX-like compounds such as dinoflagellate [85], shellfish [86], and parrotfish [87]. To date, several analogs of PLTXs and four groups of PLTX-like compounds with approximately 20 components have been identified [20]. A 42-hydroxy-PLTX analog is frequently mentioned, which has been isolated in several PLTX-containing organisms, but other analogs have been studied little because of their complex structures [88,89]. PLTXs can cause severe physiological reactions in humans, including cardiovascular and neurological symptoms, often leading to a threat to life [62,90].

## 4. Toxin Distribution in the Anatomy of Toxic Crabs

TTX, PST, and PLTXs accumulate in the whole body of crabs, including appendages, the carapace, and the viscera [4,43]. However, the toxicity of the toxic crabs varies from site to site. Hashimoto et al. reported that the appendages are usually more toxic than other parts of the body in toxic crabs [8,41], and subsequent studies have shown that the muscles of the chelipeds and the viscera were always more toxic than other parts in almost all toxic crabs, regardless of PST and TTX concentrations [50,91]. Although there have been far fewer reports on xanthid crabs containing PLTXs than those containing PSTs or TTX, two PLTX-containing crabs reported by Yasumoto et al. exhibited the highest toxicity in their gills, viscera, and eggs [43].

## 5. Geographic Distribution and Individual Variations in Toxins in Toxic Xanthid Crabs

Understanding the geographic distribution and individual variations in toxin content and profiles among toxic xanthid crabs is essential for investigating the toxification mechanisms in toxic crabs and searching for their origins. The toxicity and specific toxin profiles of these crabs exhibit significant variability, influenced by species and habitat, even within small geographic regions [9].

### 5.1. Local and Individual Variations in Toxic Xanthid Crabs

Extensive research on the toxicity of xanthid crabs across various countries and regions has been reported. A total of 119 specimens of *Z. aeneus* collected from Ishigaki Island and Hachijo Island in Japan, and from Bonin Island and Cebu Island in the Philippines, were analyzed. The results showed no significant differences in toxicity between sexes or years, although a notable individual variation in toxicity was observed [33]. On the other hand, *A. floridus* exhibited notable variability in toxicity, which appeared to be influenced by both geographic location and the season of collection in Australia [8,51]. In Taiwan, lower toxicity levels of *A. floridus* were reported [39]. For toxin profiles, *Z. aeneus*, *A. floridus*, and *P. granulosa*, which inhabit the Ryukyu and Amami Islands of Japan, the Philippines, and Australia, displayed high levels of PSTs in their toxin profiles [33,35,51]. In contrast, specimens from the Tokara Islands, the Pacific coast of mainland Japan, Taiwan, and Cebu Island in the Philippines exhibited relatively lower toxicity levels, with TTX being the predominant toxin component [3,5,35,41,57]. These findings suggested that the toxicity and toxin profiles of xanthid crabs are susceptible to differences in prey species due to complex environmental factors, resulting in geographic diversity.

### 5.2. Small-Scale Distribution of Toxin Profiles and Toxin Concentrations

Significant variations in the toxicity and toxin composition of crabs within a small area have been found on Ishigaki Island, Okinawa Prefecture, Japan. Previous studies suggested that the PST composition of *A. floridus* has been reported to differ between two reefs separated by a passage (Sites A and B, Figure 5) and also between Site B and Kojima, a small islet immediately adjacent to the strait. *A. floridus* on Kojima islet was found to be TTX-dominant [10,33]. Recently, an examination of 53 *Z. aeneus* specimens, with GPS coordinates recorded for each one on a single reef in 2018 and 2019, revealed three distinct toxin distribution zones within this reef (Figure 5). In the northwestern zone (NW), many individuals accumulated exceptionally high concentrations of PSTs, while in the central zone (CTR), most specimens contained small amounts of TTX, with few PSTs. In contrast, the southeastern zone (SE) exhibited individuals with intermediate toxin characteristics. These findings provide important evidence that xanthid crabs do not produce their own toxins but rather ingest them from their prey or environment, even within small areas.

## 6. Human Interactions and Risks

Both TTX and PST remain highly potent even after cooking, making the consumption of xanthid crabs extremely dangerous. Furthermore, PLTX and PLTX-like compounds are potent toxins even at low concentrations [84]. These toxic crabs are easily considered toxic due to their unique patterns, but they coexist in coastal regions inhabited by edible crab species and are therefore at a high risk of accidental ingestion by humans. In some regions, especially in Southeast Asia and the Pacific Islands, these crabs are sometimes caught and consumed mistakenly [91], and several food poisoning incidents due to xanthid crab consumption have been reported [43,92,93].

The clinical symptoms from TTX and PSTs are similar and vary somewhat based on the ingestion amount. The symptoms include numbness, tingling in the limbs, gastrointestinal discomfort, muscle weakness or paralysis, and, in some cases, respiratory failure and death. These neurotoxins block Na_v_ channels, damaging neurological function and leading to serious consequences if no emergency treatment is available. However, there is currently no antidote available for either TTX or PSTs. Thus, the treatment mainly involves assisted respiration and gastric lavage until the toxins are naturally eliminated from the body [68,81]. On the other hand, PLTXs and PLTX-like toxins act by blocking the Na^+^/K^+^-ATPase channel [63]. When PLTXs and PLTX-like compounds are ingested, the most common symptom in humans is rhabdomyolysis [90], including muscle pain and generalized weakness, with additional nonspecific signs such as fever, nausea, and vomiting.

Over 20 cases of xanthid crab poisoning have been reported since the 1960s, with many cases occurring in the Ryukyu and Amami Islands of Japan [1], while other cases caused by toxic xanthid crabs have been reported across various countries and regions, including instances of PLTX poisoning [34,49,93,94,95,96,97]. These cases represent the importance of monitoring the marine environments and public health education where toxic xanthid crabs are found, in order to prevent further poisoning cases. Several old cases are described in previous studies [2,3,91]. A new intoxication incident caused by toxic xanthid crab *D. reynaudii* was first recorded in Vietnam [93].

In March 2021, a significant case of human poisoning linked to crab consumption was reported in Tinh Gia District, Thanh Hoa Province, Vietnam. Two crab specimens, later identified as *D. reynaudii*, were involved in the incident. *D. reynaudii*, a toxic xanthid crab, had not been previously considered toxic in Vietnam. Later, a toxicity analysis using a mouse bioassay revealed high toxin levels in the soft tissue (175 and 195 MU/g), indicating that the crabs were unsuitable for human consumption. A further chemical analysis detected no PSTs or PLTXs but did detect TTX and its two analogs, anhydroTTX and 4-*epi*TTX, highlighting the first evidence of TTX detection in *D. reynaudii* in Vietnam. This case highlights the importance of marine monitoring and public health education. Early identification of contaminated toxic species and raising awareness among fishers, vendors, and consumers about the risks associated with toxic cabs are needed. Public health campaigns can focus on identifying toxic species and recognizing early symptoms of poisoning to ensure that poisoning incidents do not occur again.

## 7. Hypotheses on the Origins of Toxin-Producing Organisms in Toxic Crabs

Several toxins are produced by microorganisms and are subsequently transported through the food chain to various organisms [87,98,99,100]. For example, although marine pufferfish are widely recognized as TTX-containing organisms, no TTX was detected in their bodies when cultured in clean seawater and fed a TTX-free diet [99,101]. On the other hand, PSTs are well-known toxins produced by specific microorganisms in both seawater and freshwater environments [68]. Therefore, bivalve aquaculture operations, which are particularly susceptible to PST accumulation, should be located away from areas prone to harmful algal blooms (HABs). PLTXs and PLTX-like compounds could be produced by dinoflagellates, and some recent reports have suggested that bacteria or cyanobacteria are involved in PLTX accumulation [52,102,103]. In toxic crabs, the variations in toxin profiles and concentrations even within a single reef suggest that toxins in these crabs are acquired through their diet. Unlike filter-feeding bivalves, which primarily consume plankton and easily accumulate toxins [104], xanthid crabs are omnivores feeding on a wide range of items, from sediments to fish remains. Thus, they may acquire toxins through the complex food web, making it difficult to trace the origins of these toxins [105]. Some common origin organisms of toxins or hypotheses of the origins of toxins in toxic crabs are summarized below.

### 7.1. Red Algae Jania sp.

The red algae *Jania* sp. is a type of red macroalgae, or seaweed, belonging to the family Corallinaceae and is characterized by its hard, calcareous, branching skeleton.

Saisho et al. [17] analyzed three species of xanthid crabs: *Z. aeneus*, *A. floridus*, and *D. perlata*. While *D. perlata* was initially regarded as non-toxic, unpublished data from Daigo in 1987 suggested it could possess weak toxicity [97]. The stomach content analysis of *Z. aeneus* specimens from Ishigaki Island revealed a diverse diet, including a green alga (*Codium* sp.), a brown alga (*Ectocarpus* sp.), and various red algae (*Jania* sp., *Polysiphonia* sp., *Galaxaura* sp., and *Hypnea* sp.), as well as poriferans, corals, bivalves, gastropods, and sand. Among these, *Jania* sp., *Hypnea* sp., corals, and sand were predominant in specimens collected in June 1979.

For *A. floridus*, a stomach content analysis from 1979 revealed a diet consisting of *Hypnea* sp., sand, shells, animal tissues, poriferans, and fish fragments. By 1980, it was found that the diet also comprised *Codium* sp., *Ectocarpus* sp., *Jania* sp., *Galaxaura* sp., *Hypnea* sp., annelids, animal tissues, corals, and sand in specimens collected on Ishigaki Island. Similarly, the weakly toxic *D. perlata* showed a comparable diet, with *Jania* sp., *Hypnea* sp., *Ectocarpus* sp., and corals as dominant components.

In 1979, *Jania* sp. was identified as containing GTX-group toxins, and TTX was detected in 1986 [98,99]. These findings led researchers to propose that *Jania* sp. could be a possible source of crab intoxication. However, the detected toxin concentrations in *Jania* sp. were relatively low (ppb level of anhydroTTX and a maximum of 1.5 MU/g for GTX1-3), insufficient to account for the high toxin levels observed in toxic xanthid crabs. Additionally, *Jania* sp. was absent from the stomach contents of *A. floridus* in 1979 and *Z. aeneus* in 2019 [12], but was found to dominate the stomach contents of the weakly toxic *D. perlata* specimens. Therefore, *Jania* sp. cannot be definitively identified as the primary origin of these toxins.

### 7.2. Ascidians

As described in Section 5.2 on the small-scale distribution of toxin profiles and toxin concentrations, the stomach contents of xanthid crabs were also examined in 2019.

In 2019, three crabs were collected from each of the PST-rich and PST-poor zones for microscopic observation and toxin analysis. After cleaning, a large number of ascidian spicules were observed in both zones, i.e., 1263 in the PST-rich zone and 1676 in the PST-poor zone, along with small quantities of foraminifers, sponge spicules, algae, and mollusk fragments. These findings contradict those of Saisho, who reported a minimal presence of other organisms besides ascidian spicules [25]. Due to the many shapes of spicules in morphologies, it is difficult to identify ascidian species only by spicules. However, the four types of ascidian spicules detected in the stomach contents of *Z. aeneus* could be classified into two genera, *Lissoclinum* sp. and *Trididemnum* sp., based on the previous reports [106,107,108]. It is worth noting that some ascidian species have been reported to be toxic and contain PSTs and/or TTX [67,109]. The toxin analysis of stomach contents suggested that the GTX group was detected in the stomach contents of crabs collected from the PST-rich zone, where *Lissoclinum* ascidian spicules were predominant; on the contrary, only a few *Lissoclinum* ascidian spicules were found, and no PSTs were detected in the stomach contents of crabs from the PST-poor zone. The authors also analyzed both the stomach contents and the toxin concentrations in all 25 specimens. The results show that, in many individuals, the GTX group was predominant in the stomach contents. Based on the GTX transformation results, the authors suggested that xanthid crabs ingest the GTX group from toxic prey and convert it to the STX group after accumulation. These results suggest that *Lissoclinum* ascidians may be a source of PSTs. This finding provides the second alternate toxin origin, after red algae *Jania* sp. in 1983 [110]. However, no live ascidian specimens have been collected; thus, conclusive evidence on two ascidian genera containing TTX and/or PSTs remains absent. Further studies focusing on the collection of live ascidians and the toxin analysis and stomach analysis of toxic crabs in other regions are necessary in the future.

### 7.3. Zoanthid Genus Palythoa

Palythoa is a typical genus containing PLTXs and is considered one of the original sources of PLTXs in marine ecosystems. Previous studies on investigating the toxin profiles of several marine organisms collected near Zoanthid (genus *Palythoa*) habitats have detected PLTXs, including in xanthid crabs and other soft coral species [20,98]. However, Yasumoto et al. were unable to detect any zoanthids in the stomach contents of the PLTX-containing crabs they investigated, thus not providing clarity on the source of PLTXs in xanthid crabs [43]. In addition, it seems that PLTXs in the genus *Palythoa* and another zoanthid coelenterate, *Zoanthus*, are not produced by the organisms themselves. The natural bioaccumulation of PLTXs in these species via the food web is still unknown [20,111].

### 7.4. Microorganisms

Microorganisms that produce toxins, such as bacteria, dinoflagellates, and cyanobacteria, are found globally [112,113,114]. In 1986, Noguchi et al. [115] isolated *Vibrio* sp. bacteria from the intestine of *A. floridus*, identifying TTX and anhydroTTX in cellular extracts and culture medium by using instrumental analysis. This marked the first identification of a TTX-producing bacterial strain as the symbiotic microflora within a TTX-bearing organism. Since then, TTX-producing bacteria have been isolated from various TTX-bearing organisms, such as the red algae *Jania* sp. [116] and the horseshoe crab *Carcinoscorpius rotundicauda* [117]. In 1990, Kogure et al. [118] reported significant concentrations of TTX in marine sediments, providing a new perspective on free TTX-producing bacteria as the origin organism. To date, approximately 50 TTX-producing bacterial strains have been isolated from deep-sea sediments or freshwater sediments. In addition, more than 150 TTX-producing bacterial strains have been identified, most belonging to the phylum *Proteobacteria* and class *Gammaproteobacteria* [112]. Some researchers suggest that free-living TTX-producing bacteria, rather than symbiotic ones, may be the origin source of TTX [119,120,121]. These free-living bacteria may be transferred through food chains, contributing to the presence of TTX in various organisms. However, this hypothesis is currently difficult to verify due to the challenges in culturing TTX-producing bacteria under laboratory conditions and the low concentration of TTX produced by symbiotic bacteria. Several researchers point out that the bacteria will lose TTX production ability after culture in the laboratory [112]. Further investigation on the optimal conditions and external factors for growth is needed, or other methods to detect the symbiotic bacteria, such as immunoelectron and immunofluorescent microscopies and high-throughput sequencing techniques [122,123].

PSTs are known to be produced by toxic dinoflagellates, such as the genera *Alexandrium* and *Gymnodinium*, in marine environments, and by cyanobacteria, such as the genus *Anabaena*, in freshwater [68]. On the other hand, dinoflagellates of the genus *Ostreopsis*, bacteria, and cyanobacteria also have been considered to be the origin organisms of PLTXs [102,103,111,124]. However, crabs are not plankton feeders, the lack of evidence of red tides caused by PST-producing dinoflagellates in the habitats of toxic xanthid crabs reveals that the accumulation of PSTs directly via dinoflagellates seems unlikely [2], and no microflora involved in palytoxin production has been reported from toxic xanthid crabs. Interestingly, certain cyanobacteria isolated from the tunic of ascidians may provide an alternative method of accumulating PSTs and PLTXs, given that ascidians have been identified as potential origin organisms. This hypothesis needs to be verified for further research [125,126].

## 8. Resistibility and Defenses of Toxic Crabs

TTX/PST-bearing organisms, such as toxic marine pufferfish [127], gobyfish *Yongeichthys criniger* [128], bivalves [129], and newts [130], show resistance to TTX or PSTs. In fact, toxic xanthid crabs also show extremely high resistance to both TTX and PSTs.

Several experiments have been reported on the resistance of toxic and non-toxic crabs [127,131,132]. The results suggested that *A. floridus* demonstrated approximately 500 times higher resistance to the STX group, while both *A. floridus* and *Z. aeneus* exhibited about 5000 times higher resistance to the GTX group and 500 times higher resistance to TTX compared to non-toxic crabs. Furthermore, Daigo et al. [133] investigated the stability of neurons in the response to TTX and PSTs. They examined the effects of various concentrations of PST/TTX solutions on the action potentials of neuron fibers of test crabs, including *Z. aeneus*, *A. floridus*, *D. perlata*, and some non-toxic crabs and lobsters. The results showed that toxic crabs *Z. aeneus* and *A. floridus* showed approximately 1000 times more resistance to TTX and PSTs than non-toxic crabs and lobsters. In addition, the weakly toxic crab *D. perlata* also demonstrated resistance to these toxins. Based on these findings, it can be suggested that toxic crabs, like other toxin-bearing organisms, have evolved resistance mechanisms to TTX and PSTs, enabling them to selectively accumulate these toxins. This ability may also play an important role, allowing the crabs to use these toxins as a defense mechanism against predators. In contrast, the resistance to the toxins of PLTX-containing organisms remains unclear. It is speculated that these organisms may exhibit resistance or tolerance to PLTXs, which could help them accumulate PLTXs and/or PLTX-like compounds more effectively [20].

The mechanism of TTX and PST resistance in toxic organisms entails the resistance of Na_v_ channels to TTX/PSTs and TTX/PST-binding proteins [68,75]. In toxic organisms, mutations in the P-loop regions of the skeletal muscle Na_v_1.4 channels have been identified, which can reduce the binding affinity for TTX and/or PSTs. This phenomenon has been observed in marine organisms such as pufferfish and newts, which exhibit reduced sensitivity to TTX, as well as in bivalves, which show binding affinity for PSTs [68,134]. On the other hand, Yotsu-Yamashita et al. elucidated the primary structure and characteristics of the TTX-binding protein from the plasma of *Takifugu pardalis* and named it pufferfish saxitoxin and tetrodotoxin-binding protein (PSTBP) [135], and PST-binding proteins have also been identified from several marine organisms, including oysters [136], cockles [137], and shore crabs [138]. Llewellyn et al. [139] found a STX-binding factor in the hemolymph of several species of xanthid crabs, including toxic species such as *L. pictor* and *A. tomentosus* and non-toxic species such as *Liomera tristis* and *Chlorodiella nigra*. Interestingly, although there are no reports of purified TTX-binding protein obtained from toxic xanthid crabs, several kinds of TTX-neutralizing hemolymph proteins have been extracted from non-toxic shore crabs such as *Carcinoscorpius rotundicauda* and *Hemigrapsus sanguineus*, and their resistance to TTX is higher than that of other common organisms [138,140].

The above findings suggest that the high resistibility of toxic xanthid crabs enables them to accumulate higher concentrations of toxins compared to other organisms. This resistibility appears to involve mechanisms such as reducing the sensitivity of their nervous system to TTX and PSTs and utilizing a hemolymph factor that binds and neutralizes these toxins.

## 9. Ecological Role of Toxic Xanthid Crabs

It has been presumed that the toxic crabs use their toxins for defense [42]. Toxic crabs move slowly and seem to show little aggressiveness compared to other crabs, such as the “intermediate” toxic crab *D. perlata*, which is usually found on the same reef. They can be easily caught by hand, and they close their claws very slowly. However, studies have shown that toxic xanthid crabs can release their toxin when their carapace is mildly stimulated. The maximum amount of toxin released by toxic xanthid crabs ranges from 162 MU to 2635 MU for a total of 56 days, indicating that they use toxins to protect themselves rather than for physical defense [141].

Toxic xanthid crabs have been studied for over 60 years, and the potential toxin-origin organisms and their symbiotic relationships with bacteria have been discussed earlier in this review. However, only a few studies have reported their predators. Noguchi et al. [2] hypothesized that carnivorous octopuses, such as blue-ringed octopuses, may prey on toxic xanthid crabs. Blue-ringed octopuses are known to possess TTX in their posterior salivary glands [142]; however, no direct evidence is available that confirms that these toxic octopuses prey on toxic xanthid crabs, although indirect evidence was reported in 2023. Zhang et al. [42] reported similar TTX concentrations and profiles in *A. floridus* and blue-lined octopus *Hapalochlaena* cf. *fasciata* collected from the same location. This study represents the first report of toxic xanthid crabs and blue-lined octopuses collected from the same site at the same time, investigating the potential ecological relationship between these two toxic species. The results showed that, in the extraction of *A. floridus*, major components such as TTX and 11-*nor*TTX-6(*S*)-ol, with 4-*epi*TTX, 11-deoxyTTX, and 4,9-anhydroTTX as minor components, were identified. This toxin profile is similar to the one from the posterior salivary glands of octopuses collected from the same site, especially the major component 11-*nor*TTX-6(*S*)-ol. Based on these findings and the fact that octopuses are known to prey on crabs [143,144], the authors hypothesized that a predator–prey relationship may exist between these two organisms, at least in this region, where these two organisms were found together. Contrarily, octopuses from the genus *Hapalochlaena* possess a range of toxins that they use for both offense and defense [143,145,146]. Thus, the authors also hypothesized that these octopuses may attack xanthid crabs using other toxins, as the TTX resistance observed in *A. floridus* makes it difficult to paralyze them with TTX.

## 10. Discussion

Toxic xanthid crabs have been observed on coral reefs in tropical and subtropical coastal environments for over half a century. However, the origin of crab toxins and toxification mechanisms remain unknown. One of the difficulties in elucidating the toxification mechanism in toxic crabs is that few organisms have been found to be the source of crab toxins. Initially, only three species of xanthid crabs (*Z. aeneus*, *A. floridus*, and *P. granulosa*) were identified to be toxic in Japan [3]. Today, more than ten species are known to be toxic (Table 1), but *Z. aeneus* and *A. floridus* are still the most commonly studied species, accounting for 80% of all investigated specimens among toxic species, as shown in Figure 6, and over 70% of individuals of these species were found to be toxic.

There have been no reports of PLTX-containing xanthid crabs in recent years, but crabs containing TTXs and PSTs have been reported [42,93]. Global climate change associated with increased atmospheric CO_2_ concentrations and warmer water temperatures is thought to increase the abundance and promote the widespread distribution of TTX and PSTs in the marine environment [147]. This phenomenon may affect not only the presence of free TTX and PSTs but also the distribution of PST- and TTX-bearing organisms. PST-producing dinoflagellates are generally found at higher latitudes, and toxic xanthid crabs and octopuses have been collected in South Korea [148,149,150]. As the distribution of toxic prey changes or expands, non-toxic species with toxin resistance, such as *D. perlata*, a xanthid crab species, may become toxic. It is likely that crabs previously considered non-toxic and/or low-toxicity species could be transformed into toxic and/or highly toxic species. Similar risks exist for PLTX-bearing prey such as *Ostreopsis* (dinoflagellates) and *Palythoa* (soft corals), and they need to be monitored, as climate change may increase the risk of toxification of crabs.

Initially, the mouse bioassay (MBA) was widely used to detect toxins, but it was not adequately sensitive and did not reveal the toxin composition. Various instrumental analytical methods have been developed for toxin composition analysis. HPLC is now commonly used for PST analysis, and HPLC fluorescence detection (HPLC-FLD) with post-column, LC-MS, ultra-high-performance liquid chromatography (UHPLC)–tandem mass spectrometry (UHPLC-MS/MS), etc., are highly sensitive analytical methods suitable for the qualitative and quantitative analyses of TTX, PLTXs, and PSTs [20,151,152,153]. These instrumental analytical methods are now adequately developed to provide highly sensitive, accurate, and detailed information on toxin concentrations and toxin components in toxic organisms. Thus, they have become one of the reliable tools for deciphering the origin of crab toxins by obtaining detailed toxin profiles and other information from small amounts of sample.

To date, the origin organisms of crab toxins remain poorly understood, except for the toxification mechanism of PSTs that has been identified in two shore crabs, *Thalamita acutidens* and *Charybdis japonica* [154]. In the past, attempts have been made to trace the origin of the toxin by observing the stomach contents. However, stomach content analysis has limitations. First, xanthid crabs are omnivorous, and stomach contents vary from region to region and from individual to individual due to differences in the prey fauna of each habitat. In addition, the stomach contents of captured crabs often have little soft tissue remaining due to digestion, and hard calcareous tissues such as shells and bone fragments are often prominent. Furthermore, while it is possible to assess toxicity in the entire stomach content, it is difficult to analyze individual toxins because only very small fragments of each organism remain in the stomach contents. It is also difficult to identify the toxic prey species because the toxin has already leached into the stomach and contaminated the entire stomach content. in recent years, additional attempts have been made to identify the species through genetic analysis [155]. While modern genomic and metagenomic tools have not yet been applied to the study of toxic xanthid crabs, advanced techniques such as 16S rRNA sequencing using high-throughput sequencing technologies have been successfully used in research on puffer fish [123,156]. In addition, genomic mining tools such as antibiotics and secondary metabolite analysis shell (antiSMASH) were also used to research the TTX-producing bacteria *Bacillus* sp. 1839 [157].

Although the toxins in toxic xanthid crabs are thought to play a defensive role, specific predators of these crabs have rarely been identified. In Nagasaki, toxic octopuses and *A. floridus* found in the same habitat had similar TTX profiles, suggesting that the octopuses may be feeding on the crabs.

## 11. Conclusions and Future Perspective

Toxic xanthid crabs are creatures that exhibit a unique toxin transformation. However, their toxicity and their mechanisms of toxin transportation within their bodies and toxin transformation through the food web are still not well understood. Recent studies provide insights into the toxin-origin organisms and potential predators of toxic xanthid crabs. However, continued investigations are essential to comprehensively understand their ecological roles and their toxin accumulation, transportation, and transformation processes. Moreover, marine monitoring and public health education are needed to prevent the risks associated with these toxic crabs.

## Figures and Tables

**Figure 1 toxins-17-00228-f001:**
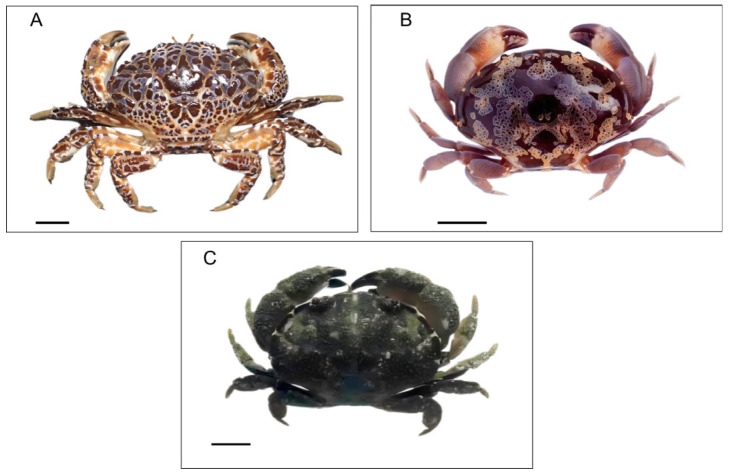
Photos of the typical toxic xanthid crabs: (**A**) *Zosimus aeneus*, (**B**) *Atergatis floridus*, and (**C**) *Platypodia granulosa*. Scale bar: 1 cm.

**Figure 2 toxins-17-00228-f002:**
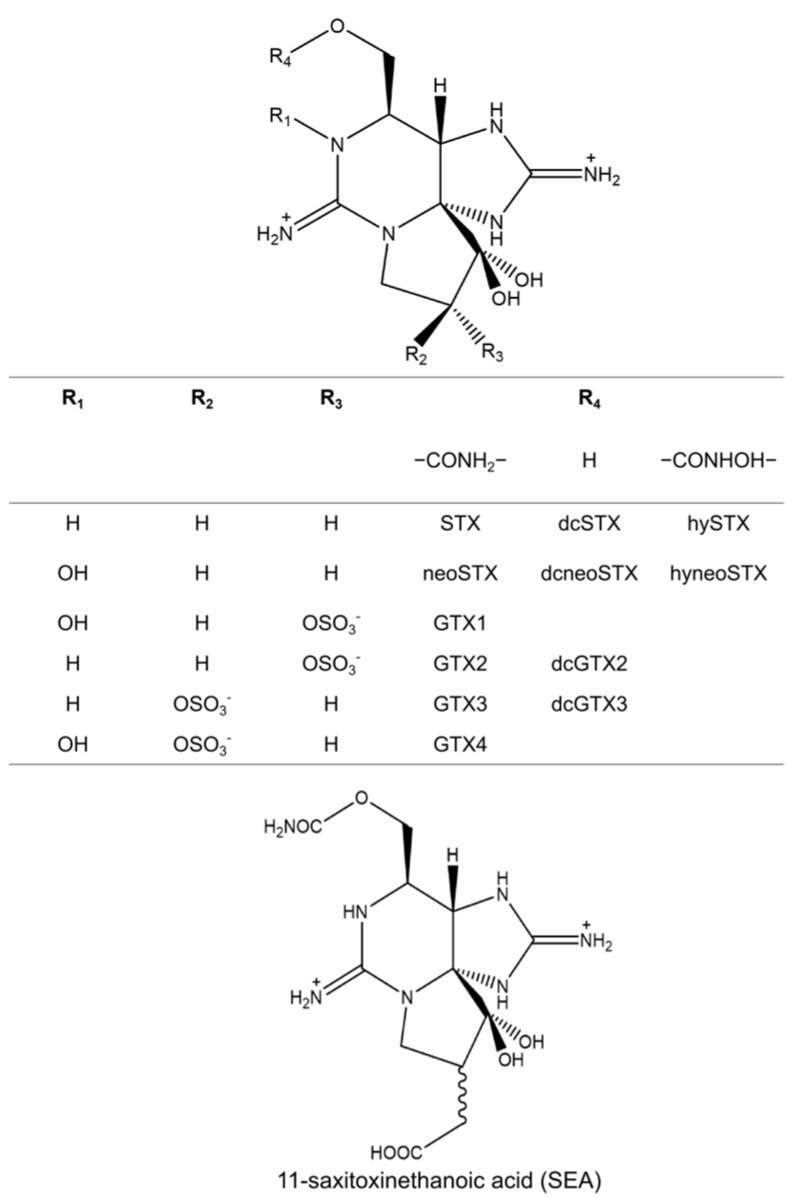
The structures of PST and PST analogs usually detected in toxic xanthid crabs. STX: saxitoxin; GTX: gonyautoxin; dc: decarbamoyl; hy: carbamoyl-*N*-hydroxy; hyneo: carbamoyl-*N*-hydroxyneo.

**Figure 3 toxins-17-00228-f003:**
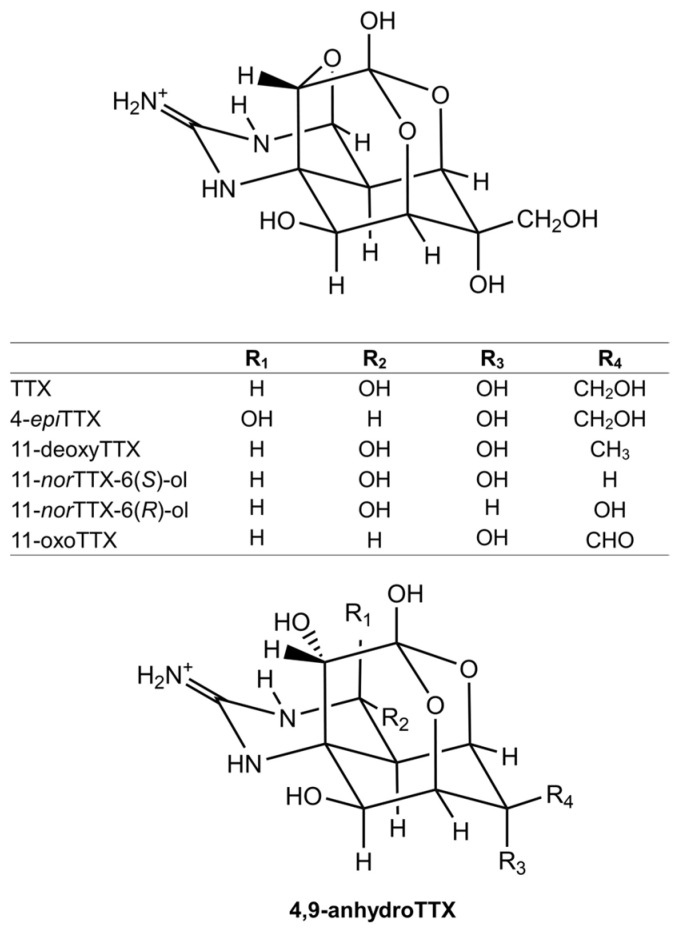
The structures of TTX and TTX analogs usually detected in toxic xanthid crabs.

**Figure 4 toxins-17-00228-f004:**
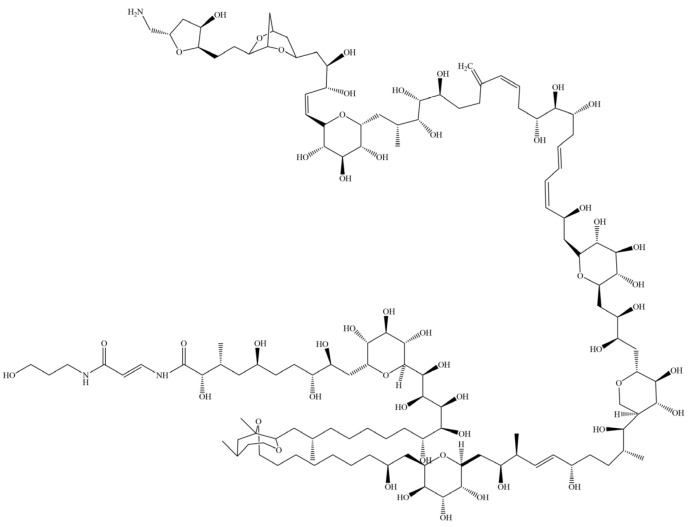
Structure of palytoxin.

**Figure 5 toxins-17-00228-f005:**
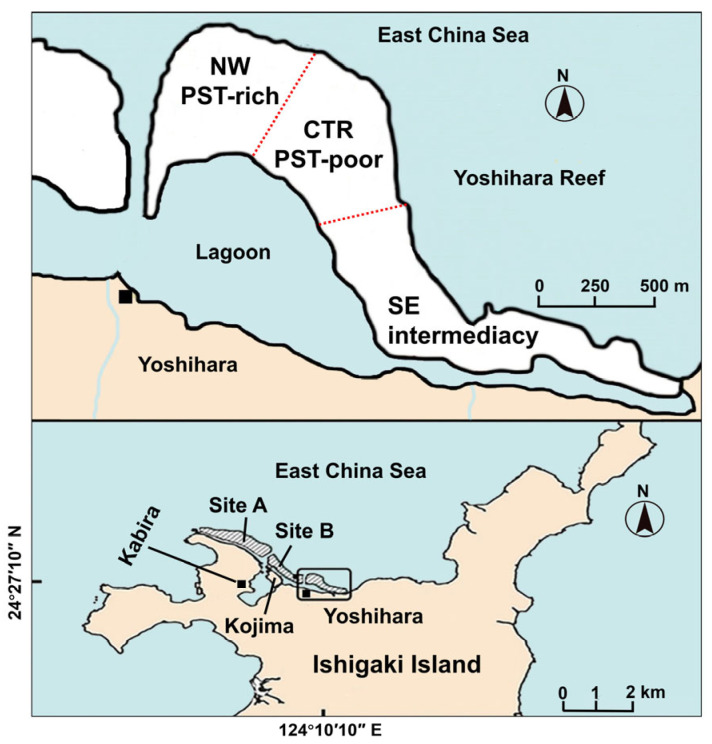
Maps of the small-scale distribution of toxin profiles and toxin concentrations: Site A and Site B from Asakawa et al. [41], and three zones on Yoshihara Reef from Zhang et al. [9].

**Figure 6 toxins-17-00228-f006:**
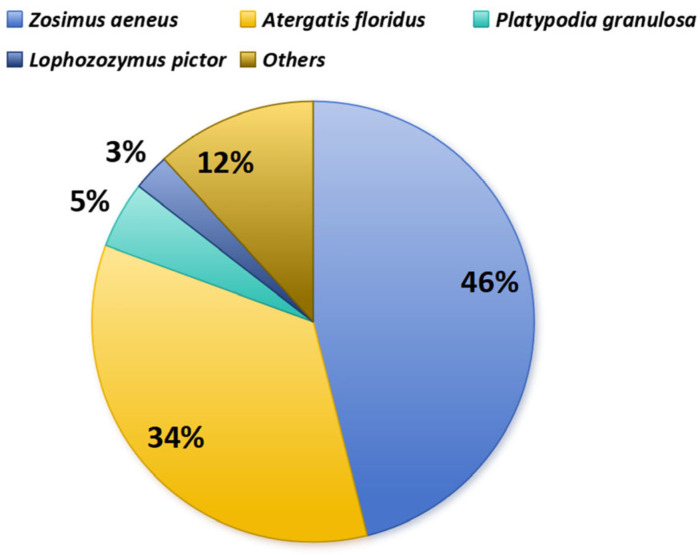
Frequency of toxic xanthid crab investigations.

## Data Availability

No new data were created or analyzed in this study.

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
