# Peer review of "Toxin Accumulation, Distribution, and Sources of Toxic Xanthid Crabs"

_toxins, 2025, doi:10.3390/toxins17050228_

Round 1
Reviewer 1 Report
Comments and Suggestions for Authors
- The article mentions gut bacteria as a potential toxin source—are there specific strains identified in A. floridus that could be further explored for biosynthetic pathways?
- Have any molecular or genetic studies been conducted to understand the mutations in sodium channels (Nav) that confer resistance to TTX and PSTs in xanthid crabs?
- Is there evidence of vertical or horizontal transmission of toxin-producing microbes within crab populations or between generations?
- How might changing ocean temperatures and pollution impact the microbial communities responsible for toxin biosynthesis in these crabs?
- Did the review find any notable shifts in crab toxicity levels or distribution over recent decades that could be linked to climate change?
- Are there currently any regional or international policies regulating the harvesting or sale of xanthid crabs for food? If not, would you recommend any?
- What are the biggest challenges in visually distinguishing toxic xanthid crab species from their non-toxic counterparts, and could AI or image recognition be leveraged for public safety?
- What are the main obstacles in pinpointing the exact origin of the toxins in xanthid crabs—environmental variability, lack of molecular tools, or insufficient microbial profiling?
- Given the current advances in metagenomics and metabolomics, what future research strategies would you propose to identify and characterize unknown toxin sources in these crabs?

Reviewer 2 Report
Comments and Suggestions for Authors
Dear Editor, I have read in detail the article entitled “Toxin Accumulation, Distribution, and Sources of Toxic Xanthid Crabs”.
The article is very interesting and provides a summary of the global research on toxic xanthid crabs, containing new findings and hypotheses on the processes of toxin accumulation in and the origins of these crabs.
My minor suggestions:
a) Include figure to more adequately illustrate the authors' description in point 2.
b) Please revise the acronym used in the text for palytoxin. They tend to confuse with another group of toxins.
c) Please describe figures 1 and 2 more adequately.
Reviewer 3 Report
Comments and Suggestions for Authors
This manuscript provides a comprehensive review of toxic xanthid crabs, including their toxin accumulation, anatomical distribution, geographic prevalence, suspected toxin sources, and ecological significance. It systematically compiles historical and recent data on various species, toxins (TTX, PSTs, PTXs), their analogs, and suspected origins, supported by extensive references and tables. The paper also discusses resistance mechanisms, potential predators, and risks to human health from crab consumption.
The manuscript is well-researched, logically organized, and offers valuable insights into a field that remains underexplored despite its public health implications. The integration of molecular, ecological, and biochemical perspectives makes this an informative and potentially influential contribution to the
Major Comments
- While the review is focused on xanthid crabs, some content (e.g., PST mechanisms in bivalves or toxin-producing algae) could be more succinct to maintain focus. A clearer distinction between xanthid-specific insights and general marine toxinology would improve readability.
- The section on stomach content as a method for identifying toxin sources is very informative. However, the manuscript would benefit from a critical evaluation of the limitations of this method (e.g., digestion state, environmental contamination).
- Improve the introduction by providing examples of devices for detecting toxins, such as aflatoxin, ochratoxin, and algal toxins (org/10.3389/fchem.2021.626630). Cite them
- The discussion of symbiotic and free-living microbial toxin sources is intriguing. It would be helpful to more explicitly highlight the challenges of confirming these origins and propose specific experimental strategies to resolve them.
- The inclusion of the 2021 Vietnam poisoning incident adds timeliness. Consider expanding on its implications for current regulatory and educational efforts.
- Figures: Ensure that figures (e.g., structures of toxins and distribution maps) are high-resolution and formatted according to the journal's style.
- References: Some older studies (e.g., from the 1960s–1980s) are foundational, but integrating more recent findings (post-2020) where available could further enhance relevance.
- Language and Style: The English is generally clear, though minor grammatical edits would improve flow (e.g., “toxification” might be better phrased as “toxin accumulation process” or “acquisition” in some contexts).
- What modern genomic or metagenomic tools could be used to identify microbial toxin producers in xanthid crab microbiomes?
- Could isotopic tracing be used to track dietary toxin acquisition in crabs?
- Are there ongoing surveillance efforts in regions where toxic crabs co-exist with edible species to prevent misidentification?
Minor Editing
Round 2
Reviewer 3 Report
Comments and Suggestions for Authors
Minor Revisions
